# Comparison of Microglial Morphology and Function in Primary Cerebellar Cell Cultures on Collagen and Collagen-Mimetic Hydrogels

**DOI:** 10.3390/biomedicines10051023

**Published:** 2022-04-29

**Authors:** Zbigniev Balion, Nataša Svirskienė, Gytis Svirskis, Hermanas Inokaitis, Vytautas Cėpla, Artūras Ulčinas, Tadas Jelinskas, Romuald Eimont, Neringa Paužienė, Ramūnas Valiokas, Aistė Jekabsone

**Affiliations:** 1Laboratory of Pharmaceutical Sciences, Institute of Pharmaceutical Technologies, Faculty of Pharmacy, Lithuanian University of Health Sciences, Sukilėlių Ave. 13, LT-50162 Kaunas, Lithuania; zbigniev.balion@lsmuni.lt; 2Laboratory of Biochemistry, Neuroscience Institute, Lithuanian University of Health Sciences, Eivenių Str. 4, LT-50161 Kaunas, Lithuania; 3Laboratory of Neurophysiology, Neuroscience Institute, Lithuanian University of Health Sciences, Eivenių Str. 4, LT-50161 Kaunas, Lithuania; natasa.svirskiene@lsmuni.lt (N.S.); gytis.svirskis@lsmuni.lt (G.S.); 4Institute of Anatomy, Faculty of Medicine, Lithuanian University of Health Sciences, Mickeviciaus 9, LT-43074 Kaunas, Lithuania; hermanas.inokaitis@lsmuni.lt (H.I.); neringa.pauziene@lsmuni.lt (N.P.); 5Ferentis UAB, Savanorių 231, LT-02300 Vilnius, Lithuania; vytautas@ferentis.eu (V.C.); tadas@ferentis.eu (T.J.); romuald@ferentis.eu (R.E.); ramunas@ferentis.eu (R.V.); 6Department of Nanoengineering, Center for Physical Sciences and Technology, Savanorių 231, LT-02300 Vilnius, Lithuania; ulcinas@ftmc.lt; 7Laboratory of Preclinical Drug Investigation, Institute of Cardiology, Lithuanian University of Health Sciences, Sukileliu Ave. 15, LT-50162 Kaunas, Lithuania

**Keywords:** collagen-like peptide, crosslinked collagen, hydrogel, TEM, cryo-FIB/SEM, microglia, Ca^2+^ oscillations

## Abstract

Neuronal-glial cell cultures are usually grown attached to or encapsulated in an adhesive environment as evenly distributed networks lacking tissue-like cell density, organization and morphology. In such cultures, microglia have activated amoeboid morphology and do not display extended and intensively branched processes characteristic of the ramified tissue microglia. We have recently described self-assembling functional cerebellar organoids promoted by hydrogels containing collagen-like peptides (CLPs) conjugated to a polyethylene glycol (PEG) core. Spontaneous neuronal activity was accompanied by changes in the microglial morphology and behavior, suggesting the cells might play an essential role in forming the functional neuronal networks in response to the peptide signalling. The present study examines microglial cell morphology and function in cerebellar cell organoid cultures on CLP-PEG hydrogels and compares them to the cultures on crosslinked collagen hydrogels of similar elastomechanical properties. Material characterization suggested more expressed fibril orientation and denser packaging in crosslinked collagen than CLP-PEG. However, CLP-PEG promoted a significantly higher microglial motility (determined by time-lapse imaging) accompanied by highly diverse morphology including the ramified (brightfield and confocal microscopy), more active Ca^2+^ signalling (intracellular Ca^2+^ fluorescence recordings), and moderate inflammatory cytokine level (ELISA). On the contrary, on the collagen hydrogels, microglial cells were significantly less active and mostly round-shaped. In addition, the latter hydrogels did not support the neuron synaptic activity. Our findings indicate that the synthetic CLP-PEG hydrogels ensure more tissue-like microglial morphology, motility, and function than the crosslinked collagen substrates.

## 1. Introduction

The current strategies for 2D and 3D neural cell culture matrices are based on applying a high density of molecules promoting adhesion-mediated neuronal-glial differentiation and neuritogenesis: for example, polylysine, polyornithine, laminin fibronectin, collagen and others [1]. Such cultures consist of evenly distributed neuronal and glial networks that are easy to visualize and monitor; however, the cellular density and morphology are far from those in the native tissue. Microglial cells are usually in a highly inflammatory activated state characterized by a large body and short processes. They are different from ramified microglia in the brain tissue forming a small body and highly branched extensions responsible for synaptic scanning and pruning [2,3]. Although well-attached cells are convenient for monitoring throughout the experiment, they might lack the ability to move and interact and, thus, might prevent the formation of tissue-like intercellular connections. We have recently reported on the effect of a semi-adhesive, synthetic matrix made from collagen-like peptides conjugated to polyethylene glycol (CLP-PEG) and found it to promote functional organoid formation from isolated primary rat cerebellar cells [4,5]. The cultures on such hydrogels were characterized by spontaneous organizing primary cerebellar cells into specifically connected tissue-like clusters with close interaction between neurons, astrocytes and microglia, generating spontaneous neuronal Ca^2+^ signals. We found that CLP-PEG hydrogels caused increased numbers of microglial cells in proximity to the neuronal clusters, suggesting they might play a role in developing the synaptically active structures. Microglia are the main resident immune cells in the brain [6], and calcium signalling is vital for their function [6,7], including cytokine release [8], P2X receptor trafficking and diffusion [9]. Calcium-dependent fluorescence in vivo imaging has demonstrated that microglial cells have spontaneous Ca^2+^ signalling, but it is not frequently observed in a quiescent state [10,11,12]. Differences in the Ca^2+^ signalling pattern usually report changes in the microglial function. For example, the rate of the signals increases after local neuronal tissue injury [11], and signal frequency and duration increase in a microglial fraction-dependent manner during ageing [13]. In addition, neuronal activity also induces increased microglial process Ca^2+^ signalling related to the process extension [14]. Moreover, neuronal-derived signals influence the maturation of microglial cells [15]. On the other hand, microglia regulate the formation of synaptic connections [16] and neuronal activity [17] in developing and mature brains. Therefore, in this study, we have more thoroughly analyzed the morphology and function of microglial cells cultivated on the CLP-PEG hydrogels. In addition, we have compared the cerebellar cell culture development and microglial features on the synthetic CLP-PEG and crosslinked collagen hydrogel matrices.

## 2. Materials and Methods

### 2.1. Hydrogel Synthesis

The CLP-PEG hydrogel was fabricated as described in [5,18]. Briefly, the synthetic peptide (Cys-Gly-(Pro-Lys-Gly)_4_(Pro-Hyp-Gly)_4_(Asp-Hyp-Gly)_4_) were obtained from UAB Ferentis (Vilnius, Lithuania) and it was conjugated to a 40 kDa, 8-arm PEG-maleimide (hexa-glycerol core (Creative PEGWorks, NC, USA or JenKem, TX, USA). Next, 12% (*w*/*w*) CLP-PEG was crosslinked by N-hydroxysuccinimide (NHS) and N-(3-dimethylaminopropyl)-N′-ethylcarbodiimide hydrochloride (EDC) in aqueous solution using a molar equivalent for peptide-NH_2_:NHS:EDC of 1:2:2. The hydrogel was cast between two glass slides with a 500 µm-thick spacer and allowed to cure for 16 h in a humidified chamber. Similarly, collagen hydrogel was prepared from 12% (*w*/*w*) porcine collagen type I (NMP collagen PS, Nippon Meatpackers, Ibaraki, Japan) as described elsewhere [19]. The CLP-PEG hydrogels with a microwell topography were made using the same protocol, but instead of flat glass slides, the hydrogels were cast onto a polydimethylsiloxane (PDMS; Dow Corning, Midland, MI, USA) replica mold of the commercially available cellular spheroid growing plate AggreWell^TM^ 400 (STEMCELL Technologies Canada Inc., Vancouver, BC, Canada) with 400 μm size microwells (Appendix A). Finally, 6 mm hydrogel disks were cut from the fabricated sheets using a trephine (Kai Industries Co., Seki, Japan).

### 2.2. Visualization of Hydrogel Microslices

The hydrogel was immersed in eosin (500 mg eosin in 100 mL 70% ethanol) for 10–20 s and briefly washed in PBS (pH 7.4). The excess of the buffer was absorbed with filter paper. The discs were frozen at −40 °C in Thermo Scientific^TM^ Shandon^TM^ Cryomatrix^TM^ mounting medium on a cryostat working table and sliced parallel to the surface in 8 μm sections by a cryomicrotome (CryoStar NX70, Walldorf, Germany). The slices were mounted on microscope glass slides (Superfrost Plus, Menzel-Gläser, Germany) and allowed to dry at room temperature for 15–30 min. Then the slides were briefly immersed in Leica ST Ultra solvent and covered by cover medium (Leica CV Ultra) and coverslips. The slices were visualized under a Zeiss Axio Observer Z.1 light microscope (Carl Zeiss AG, Mannedorf, Germany), and images were captured by the Axio CamMRm digital camera.

### 2.3. Transmission Electron Microscopy

Collagen hydrogel and CLP-PEG discs (0.5 mm thick and 6 mm diameter) were fixed overnight at 4 °C in 2.5% glutaraldehyde in 0.1 M phosphate buffer (pH 7.4). The tissue samples were post-fixed for 2 h with 1% osmium tetroxide in 0.1 M phosphate buffer (pH 7.4), dehydrated through a graded ethanol series and embedded into a mixture of resins Epon 812 and Araldite (Sigma-Aldrich, Steinheim, Germany) employing an automated tissue processor LYNX II (EMS, Hatfield, PA, USA). Ultrathin sections (70 nm) were cut with the ultra-microtome Leica EM UC7 (Leica Mikrosysteme Handelsges.m.b.H., Vienna, Austria), mounted on 600 mesh thin bar support nickel grids (Agar Scientific, Essex, UK), and stained with uranyl acetate and lead citrate with an automated TEM stainer Q-3000SC (RMC, Tucson, AZ, USA). Finally, the ultrathin sections were investigated at 100 kV with a transmission electron microscope Tecnai BioTwin Spirit G2 (FEI, Eindhoven, The Netherlands). Electron microscopy images were taken with a bottom-mounted 16 Megapixel TEM CCD camera Eagle 4K, employing its specific software TIA (FEI, Eindhoven, The Netherlands).

### 2.4. Focused Ion Beam Scanning Electron Microscopy

For SEM imaging, we employed a Zeiss Crossbeam 1540 FIB/SEM microscope (Carl Zeiss AG, Mannedorf, Germany) equipped with a Baltec VCT100 cryo stage and cryo transfer system. In this microscope, a sample can be processed under cryo conditions (temperature < −140 °C) and at room temperature. First, imaging of frozen samples was conducted, and afterwards, samples were prepared for room temperature imaging by freeze-drying. The samples were visualised by Everhart–Thornley and in-lens secondary electron detectors. Images were recorded with an acceleration voltage of 1.8 kV in a 1024 × 768 pixels format, at integration times between 15 μs and 45 μs per pixel.

### 2.5. Atomic Force Microscopy

Surface topography images of CLP-PEG and collagen hydrogels were acquired using a NanoWizard 3 JPK atomic force microscope (AFM) equipped with the MLCT-A (Bruker) AFM probes in PBS pH = 7.4 buffer solution. Imaging was carried out using the quantitative imaging (QI) mode at the 0.4–5 nN force setpoint; three different regions of each sample were scanned. The scan size ranged from 10 × 10 to 2.5 × 2.5 μm^2^ at the 256 × 256 pixel resolution. Three samples were analyzed for each hydrogel type.

The AFM nanoindentation technique setup used for the elastic (Young) modulus measurement of the collagen hydrogel samples was applied following the earlier published protocol [5]. In total, measurements of sixteen samples from four different batches were performed.

### 2.6. Neuronal-Glial Cell Culture

All experimental procedures were performed according to the Guide for the Care and Use of Laboratory Animals. The rats were maintained at the Lithuanian University of Health Sciences animal facility in agreement with the Guide for the Care and Use of Laboratory Rats. The experimental routine was performed as stated previously [5]. Briefly, the cerebella from 5–7-day-old Wistar rats were minced and incubated in Versene solution (1:5000; Gibco, Thermo Fisher Scientific, Waltham, MA, USA) at 37 °C for 5 min, then repeatedly triturated with a Pasteur pipette and centrifuged at 270× *g* for 5 min. Next, the sedimented cells were resuspended in a DMEM medium with Glutamax (Thermo Fisher Scientific) supplemented with 5% horse serum, 5% fetal calf serum, 38 mM glucose, 25 mM KCl, and antibiotic-antimycotic (Thermo Fisher Scientific). The cells were plated at a density of 0.25 × 10^6^ cells/cm^2^ in 96-well plates (VWR) on hydrogel substrates or directly into wells coated with 0.0001% poly-l-lysine. The cultures were grown for seven days in vitro (7 DIV) until being evaluated for the cell number, cell culture shape, microglial morphology, motility and Ca^2+^ signaling.

### 2.7. Immunostaining of Primary Cerebellar Cells, Cell Counting and Neurite Assessment

The cells were stained as described elsewhere [5]: the nuclei were identified by Hoechst33342 (6 µg/mL, 15 min at 37 °C), neurons—by microtubule-associated protein 2 (MAP2), astrocytes—by glial fibrillary acidic protein (GFAP). The microglial cells were stained by isolectin GS-IB_4_ from *Griffonia simplificolia*, Alexa Fluor^®^ 488 conjugate (Molecular Probes, Eugene, OR, USA). The cells in cultures were detected using laser scanning confocal microscopes: Zeiss Axio Observer LSM 700 (Carl Zeiss Microimaging Inc., Jena, Germany) and Olympus Fluoview FV1000 (Olympus Corporation, Tokyo, Japan). The cell number was evaluated according to the number of Hoechst33342-positive nuclei in the 400 × 500 μm^2^ area of the confocal micrographs. The amount of each cell type was represented as the percentage of the total cell number in the cultures. For neurite evaluation, the MAP2-positive area was calculated by the ImageJ software and expressed as the percentage of the entire image area per neuronal nucleus.

### 2.8. Evaluation of Microglial Motility

For evaluation of the microglial motility, time-lapse phase-contrast images of the cultures were taken for one hour each 30 s by the Olympus IX71 microscope (Olympus Corporation, Tokyo, Japan). The images were further processed by the ImageJ software to binary images to contrast the microglial cells compared to the other cells and background (an example of such an image before and after processing is presented in Figure 1a,b). The path of each microglial cell is indicated as a yellow track on a black/white image by ImageJ plugin TrackMate [20], see Figure 1c. The algorithm was written in Python script employing the imaging library Pillow to open the image files, extract the cell tracks and count the total number of yellow pixels per image (Figure 1d). This number was assumed as the total path length of all microglial cells in the microscopic camera field. The pixel count was converted to μm according to the image scale, and the average microglial rate was expressed in μm/h.

### 2.9. Ca^2+^-Signalling Recording

For Ca^2+^ signal measurements, we used similar methods as in the previous study [5]. Briefly, the neuronal-glial cells on day 7 in culture were loaded with the cell-permeable Ca^2+^-sensitive dye Oregon Green^TM^ 488 BAPTA-1, AM (OGB-AM, Thermo Fisher Scientific) in BrainPhys^TM^ Neuronal Medium (Stemcell Technologies, Vancouver, BC, Canada) for 15 min. BrainPhys^TM^ Neuronal Medium was used for washing extracellular dye and recording Ca^2+^ concentration. Images were acquired and analyzed with the Solis software (Andor Technology Ltd., Belfast, UK); ImageJ and custom-written algorithms were also used for analysis. The fluorescence images were acquired for 20 s with a rate of 30 Hz. In every sample, 4–6 different regions were registered. To analyze the Ca^2+^ signal strength and generation frequency, we used the relative peak fluorescence values ΔF/F, signal amplitude divided by fluorescence intensity at the signal base, only if they exceeded the 5% threshold. For analysis of the signal duration in a cell, the half-time T1/2 was measured as duration at the level of half-amplitude. In the fluorescence trace, the parameters ΔF/F and T1/2 were evaluated for the signal showing the highest amplitude.

### 2.10. Statistical Analysis

All quantitative data in the graphs are presented as means of 4–7 experiments and the standard error. The statistical significance was evaluated by the SigmaPlot v13 software by a one-way ANOVA Tukey test. For Ca^2+^ oscillation data, the normality of the fluorescence data distribution was assessed using the Shapiro–Wilk test. The statistical significance of the difference between the averages was evaluated using an independent two-sample Welch’s *t*-test for normally distributed data. The statistical significance between the cumulative probability distributions was also assessed using the Kolmogorov-Smirnov (K-S) test. All statistical analyses were conducted using procedures from the SciPy package.

## 3. Results

### 3.1. Comparison of Micro and Nanostructure of CLP-PEG and Collagen Hydrogels

The hydrogel materials used are transparent and shape-retaining; they display similar values of the elastic modulus: 144.5 ± 26.5 kPa for CLP-PEG [5] and 151.1 ± 33.4 kPa for collagen hydrogel, respectively. In both hydrogels, the bound water content was not less than 91%. Thus, the materials would be affected by water freezing and expanding during histological cryosectioning. The ultramicroscopy analysis revealed that the CLP-PEG hydrogel has a spatially isotropic structure, resulting in the formation of round-shaped pores during the bound water freezing (see the brightfield image in Figure 2). The pore diameter was variable. However, the hydrogel structure pattern was similar from slice to slice throughout the disc, indicating isotropic structural properties. On the contrary, the collagen hydrogel structure had a highly expressed spatial orientation. There was a clear tendency to break the hydrogel in elongated linear shapes suggesting they were formed by breaking the boundaries between parallel collagen fibers. A similar pattern was also in the slices from all depths of the hydrogel.

Next, we obtained small micrometre scale SEM images of the hydrogels using a cryo transfer system, allowing both imaging under cryo conditions in temperatures down to −140 °C and at room temperature. First, frozen samples were visualized under cryo conditions and afterwards, samples were prepared for room temperature imaging by freeze-drying. Fresh-frozen CLP-PEG samples demonstrate round-shaped structures with spherical expansions of the freezing water. The structures shrunk to hexagon-like concavities after drying (Figure 2, Cryo FIB-SEM). After fresh-freezing, the surface of collagen hydrogel looks relatively smooth with occasional fibrillary structures that appeared to be edges of the collagen sheets that were detached from each other by freeze-drying. A similar lamellar structure in chemically crosslinked collagen hydrogel was reported in the literature [21].

The AFM analysis of the hydrogel topography was performed in the aqueous environment at room temperature, thus minimizing the risk for any structural deformation. Note that in contrast to the above techniques, it does not involve changing the water state or water amount and allows us to examine the structure in its native form, with the presence of bound water molecules. The analysis revealed a densely packed and smooth CLP-PEG hydrogel surface topography. Similar observations were published earlier [5]. However, the topography of the porcine collagen type I hydrogel revealed randomly oriented bundles of fibrils, resulting in a higher height variation. Bundle-like surface topography is typical for chemically crosslinked collagen-based hydrogels, for example, recombinant human collagen type III [21]. The fibrils of the collagen hydrogel were even more visible in the TEM images, where they appeared to be densely packed and parallel. The sample preparation procedure for this assay involved dehydration; thus, the distance between the fibers was minimized. The TEM images show the clear difference between the structure of the CLP-PEG and collagen hydrogels in terms of the size of the structural units formed from the fibrils. In collagen hydrogels, the parallel fibrillar structures were organized in bundles that were elongated to several micrometres and more. However, in CLP-PEG, the fibril assemblies were just several nanometers in length and thickness and had a random spatial orientation.

### 3.2. The Effect of CLP-PEG and Collagen Hydrogel on Cerebellar Cell Culture Formation

After a day in culture, the cerebellar cells on both hydrogels organized themselves into clusters. They were clearly different from the cells on the poly-l-lysine-coated tissue plastic, where they are evenly spread to a single cell layer with some occasional cells on top (Figure 3).

After three days in vitro, the cell clusters on CLP-PEG had more clear borders, and most of them obtained spherical shapes. On the collagen hydrogel samples, the clusters remained, but their shape was rather irregular. After five days, the cultures on the CLP-PEG were completely organized into spheroids connected with each other by fibers. The spheroids were surrounded by round-shaped cells with relatively large bodies that are characteristic of the active microglia. The spheroids connected by a fiber network remained until day seven; the only difference was that the morphology of the cells around the spheroids (microglia) was changed from round to more diverse shapes. In the cultures seeded on plastic, the cell distribution remained unchanged during all the days of monitoring. The only changes visible in the plastic cultures were regarding the individual cell morphology. Cerebellar granule neurons gradually obtained their typical round-body shape and formed a dense neurite network. The same cerebellar cell culture formation on the CLP-PEG hydrogel and plastic was observed earlier [5]. On collagen hydrogels, the cultures more resembled those on plastic than on CLP-PEG after five days. The clusters were gone, and the cells were spread on the surface in a single cell layer. After seven days, the distribution pattern did not change; however, the total number of visible cells was significantly decreased. Note that the best visible cells in the phase contrast are cerebellar granule neurons and microglia, while the astrocytes are not easy to distinguish from the background. Thus, immunocytochemical staining is required to evaluate the cell composition in the cultures.

Visualization of neuronal, astrocytic and microglial specific markers revealed that the spheroids in the CLP-PEG cultures are composed of densely packed cerebellar granule neurons (Figure 4). The neurites of the neurons were localized together with astrocytes, and the microglial cells surrounded these neuron-astrocyte organoids. Although some microglial cells did not seem to be connected with the neuronal-astrocyte spheroids, there is a noticeable tendency to pool around the spheroids and alongside the spheroid connecting fibers. Some of the microglial bodies are significantly smaller than others, and some processes of these cells are incorporated among the neuronal bodies inside the spheroid.

On collagen hydrogel, however, the neuronal network was poor, the neurites were short, and many neuronal bodies looked shrunk compared to the bodies in the cultures on plastic. On the contrary, the astrocytic network was substantially expressed in the cultures on collagen hydrogel. The neuronal processes were localized near astrocytes, indicating communication between the cells. Differently, in the cultures on plastic, astrocytes and neurons colocalised only occasionally. There were only a few microglial cells in the collagen hydrogel and plastic-supported cultures, and the bodies of these microglia were predominantly large and round, or close to round.

The quantitative analysis of the cell numbers showed that neurons made half of the total cells in cultures on CLP-PEG, nearly 80% in the cultures on plastic, and less than 30% on the collagen hydrogel (Figure 5a). The number of astrocytes and microglia on the CLP-PEG was nearly equal. However, the proportions of the glial cells on the collagen hydrogel was utterly different, with 60% of astrocytes and only 15% of microglia. On poly-l-lysine-coated plastic, the percentages of the astrocytes and microglia were eight and sixteen percent, accordingly.

The level of neuritogenesis expressed as the average neurite area per neuron was significantly (nearly twice) higher on the CLP-PEG compared to cultures on collagen hydrogel and plastic. Unexpectedly, the parameter was not different between the collagen hydrogel and plastic cultures. Most likely, this is due to the vast number of neurons in the cultures on plastic compared to the ones on the collagen hydrogel. Such a high neuronal density determines small distances between the neurons and a relatively small area for neurite distribution.

To summarize the results of the cell culture, primary cerebellar cell development showed a completely different pattern on the CLP-PEG and collagen hydrogels. The CLP-PEG promoted the formation of cerebellar granule neuron-astrocyte spheroids connected with neurite-astrocyte fibers and surrounded by microglia, but this did not happen on the collagen hydrogels. In addition, the collagen hydrogel significantly stimulated astrocyte proliferation but did not support neurons or neuritogenesis and microglial cells as well as CLP-PEG.

### 3.3. Comparison of Microglial Morphology and Motility on CLP-PEG and Collagen Hydrogels

In addition to the increase in the overall microglial number, the CLP-PEG hydrogel also supported a diversity of the microglial morphological forms. In the cultures visualized in brightfield microscopy using a phase-contrast objective, there were visible ramified microglia with tiny bodies and extended processes, bipolar rod-shaped microglia, round microglia with multiple spiky filopodia, and flat, well-attached to the surface with round-shaped filopodia (Figure 6a). A fluorescent image of a ramified microglial cell with multiple thin branched processes on CLP-PEG hydrogel is presented in Figure 6b (left). The size and shape of the ramified microglial cells found on the CLP-PEG were similar to the microglial cell population described as responsible for synaptic pruning in developing the cerebellum [22] (Figure 6c). There were no microglial cells with such morphology in the cultures on poly-l-lysine-coated plastic or collagen hydrogel membranes. Microglia in the cultures mentioned above were polygonal or round-shaped (Figure 6a,b).

Time-lapse imaging revealed that the bodies of the ramified microglial cells remained static, but the extended processes of the cells generated scanning-around movements by protruding and retracting filopodia (Appendix A). Branched (similar to ramified) microglial cells demonstrated process outgrowth and process retraction. The round-shaped microglial cells generated similar environment-scanning filopodia movements. Some of such round cells performed the rotational scanning without relocation. Still, the other cells were scanning and slightly migrating on the CLP-PEG surface or along the neurite fibers. There were a lot of transitions from one morphological form to another ongoing in the cell cultures on the CLP-PEG. The cells were changing from round to flat or vice versa, from flat to rod-shaped, changing their location by rolling or spreading aside on the surface and dragging the rest of the body afterwards. The fastest-moving microglia was the flat-type with amoeboid-like filopodia, sometimes leaving a thin extension remaining, or in other cases, protruding the similar one. These may be two ways of starting transformation towards the ramified morphology; however, more detailed studies are needed to confirm this.

In contrast to the cultures on CLP-PEG, a few or no microglial movements were observed in the cultures on the collagen hydrogel and plastic, except the rotational scanning around the microglial body by small spiky filopodia (Appendix A). The average rate of microglial body motility evaluated between DIV5 and DIV7 was significantly higher on the CLP-PEG than collagen and plastic cultures (Figure 6d).

### 3.4. Comparison of Microglial Ca^2+^ Signalling on CLP-PEG and Collagen Hydrogels

We used calcium-dependent fluorescence dye, OGB-AM, to record and analyze the generation of spontaneous Ca^2+^ signals lasting for about ten or fewer seconds in mixed neuronal and microglial cultures grown on the CLP-PEG and collagen hydrogel surfaces, respectively (Figure 7). Previously, it was shown that the microglia cells generate spontaneous Ca^2+^ signals in vivo and in vitro [13,23,24].

Both neurons and microglia showed spontaneous signalling in cerebellar cell cultures grown on the CLP-PEG hydrogel substrates. We have previously described the neuronal activity in such cultures [5], and in the present study, we have focused on the signalling of the microglial cells. Out of all recorded microglial cells, 14.8% (59 out of 399 in three cultures) generated spontaneous Ca^2+^ signals with the relative amplitude ΔF/F = 8.9 ± 3.2% and the half-time T1/2 = 2.6 ± 1.1 s (Figure 7b). The mean generation rate was 0.095 ± 0.05 impulses per second (imp/s). Similar low spontaneous activity of microglial cells was previously observed in vitro and in vivo [13,23,24]. In the cell cultures grown on the collagen hydrogel, only microglial cells, but not neurons, generated the spontaneous Ca^2+^ signals. Microglia on the collagen hydrogels were less active compared to the cells on the CLP-PEG; only 4.8% of the recorded cells (11 out of 288 in three cultures) generated the spontaneous Ca^2+^ signals. The relative amplitude was also smaller, ΔF/F = 7.6 ± 3.2%, by 14.7% (*p* < 0.05 in K-S test), and the half-time was longer, T1/2 = 4.4 ± 1.5 s, by 69.7% (*p* < 0.01 *t*-test) comparing to the cells on the CLP-PEG hydrogels (Figure 2). We did not calculate the mean generation rate as only one Ca^2+^ signal was observed during the 20 s recording in all active microglial cells. This indicates that the average signalling rate in the remaining cells was less than 0.05 impulses per second (imp/s). Accordingly, the rate was more than two times less than in the microglial cells grown on the CLP-PEG hydrogel membranes.

### 3.5. Cerebellar Cell Cultures in CLP-PEG Hydrogel Microwells

The above-described data confirm that primary cerebellar neuronal-glial cells on the CLP-PEG organize into 3D or 2.5D spheroids or clusters with spontaneously active neurons and glia. Still, such cellular aggregates are of varying sizes, and this might be an obstacle for standardizing the organoids for further disease modelling and drug testing applications. Therefore, we have prototyped microscopic cavity (microwell) features in the hydrogel blocks by employing the regular molding process, aiming to optimize the format towards more standardized cell cultures. We expected that such a format would evenly distribute the cells into the microwells when plating, and control the size of a spherical unit in a single microwell by varying the cell density.

Indeed, seven days after plating on the CLP-PEG hydrogel surfaces presenting the microwells, the neuronal-glial cells formed isolated spheres of a similar diameter (Figure 8a). The neurites of the neuronal cells were incorporated into the spheroids but did not extend on/into the hydrogel itself (Figure 8b). The size of the spheroids was directly dependent on the number of the cells seeded: 0.1 × 10^6^, 0.2 × 10^6^ and 0.4 × 10^6^ cells per well caused spheroids of 30.45 ± 5.59, 70.40 ± 6.32, and 115.35 ± 7.51 μm in diameter, respectively. Thus, microwell-shaped CLP-PEG provides a matrix of cerebellar organoids of desired defined sizes. However, the present format does not support connections between the individual organoids.

## 4. Discussion

Recently, a lot of effort has been made to develop more in vivo-relevant models for basic research, preclinical testing and regenerative medicine purposes. As a result, various scaffold-free and scaffold-based techniques to produce 3D and 2.5D cell cultures with certain tissue-mimicking qualities have emerged. However, due to the diversity of the design assays and the purposes they are developed and used for, it is not straightforward to classify them according to one particular parameter. However, not all in vivo-relevant systems are easy to use and analyze. Figure 9 shows an attempt of arbitrary grouping according to the in vivo relevance and time, effort-cost, and analytical and repeatability efficiency. The scheme estimates the capacity to match in vivo relevance and the feasibility/sophistication of the available cell culture tools.

While the plastic-made tools for the cell cultures prevail as the most common, it is well known they poorly reflect the cellular environment of the native tissue. Scaffolds for 3D cell cultures such as emulsion-templated polymers (Alvetex^®^), spun/woven materials and nanocomposites resemble the natural tissue structure in terms of the spatial distribution of the cells [25]. Still, they differ from the extracellular matrix (ECM) in their mechanical properties and chemistry. Furthermore, they limit the analytical capacity as they are not optically transparent. Scaffold-free techniques such as hanging drop or agarose microwells (as supported by MicroTissues^®^) are widely used for spheroid or other organoid production from isolated cells pooled together and kept close for making contacts. Such organoids match the 3D distribution requirement. However, they differ from the naturally developed tissues due to the lack of naturally produced ECM, tissue-relevant cellular composition and architecture, and, subsequently, function.

On the other hand, the emerging organ-on-a-chip systems designed from human stem cells or human inducible pluripotent stem cells for more efficient clinical testing aim to fill the relevance gap between the animal models and humans. These models combine organotypic chemical, mechanical, spatial properties and fluid flow conditions and are adjusted for non-invasive analytical techniques to preserve the system homeostasis while monitoring. However, the designs are still relatively complex and expensive, partially as a result of the special equipment needed for system maintenance and analytical monitoring. Embryoids usually made from the stem or inducible pluripotent stem cells are multilayer bodies comprising three primary germ layers (ectoderm, mesoderm and endoderm) equivalents. Organoids are more specific embryoids developed into organ-resembling structures, like the brain [26,27,28] or liver [29,30]. The generation of such systems is valuable for developmental process studies and adds knowledge about regenerative medicine strategies; however, they are complex, expensive and difficult to reproduce and analyze.

Hydrogels have also become common in cell culture work as they offer ECM mimetics in terms of the chemical structure and water content. Their mechanical properties can be selected for nearly all tissue types. Hydrogels are used for cell mixing and encapsulation or for surface-supported cell culture. However, they are mainly available as materials for “make yourself” solutions (see Figure 9), and the user has to adopt them for the particular cell culture format, determine hydrogel concentration (varying stiffness), and change the composition by mixing in molecules of interest. Despite offering such a wide range of possibilities, the typical commercial hydrogel types require particular development of the assay/format, which might cause risk for lab-to-lab reproducibility of the results. On the practical level, the preparation procedure often requires effort and is time-consuming: for example, some hydrogels (e.g., Corning^®^ Matrigel^®^) must be prepared at temperatures below 10 °C to prevent gelation; they must be handled with care to avoid bubbles and damage of the gel surface. Furthermore, imaging cells in hydrogels has certain limitations as not all hydrogels are transparent enough.

In the present study, we demonstrate the potential of self-supporting synthetic, ECM-mimetic hydrogels. The CLP-PEG hydrogel membranes can be readily used in multiwell plates. The inserts are robust enough for handling and transfer; the material itself is optically clear: the light transmission of a 12% CLP-PEG hydrogel is 92.4 ± 0.95% and the backscatter 0.90 ± 0.17% [4]. Such optical properties are beneficial for visualizing both living and fixed cells and applying colourimetric and fluorometric evaluation at any experimental point. The above-mentioned qualities make the inserts compatible with most analytical assays developed for cell culture in conventional labware.

The neural tissue is relatively challenging to mimic due to its inherent architectural and functional complexity. Therefore, the gap between the in vivo relevance and easily reproducible high-throughput screening-compatible models for this tissue is even more prominent. Most neural cell culture models are composed of neurons or neurons and astrocytes and do not include microglial cells. The role of microglia in the development and maintenance of neural tissue has been underestimated for a long time as these cells were thought to play their part only under pathological conditions. However, the recent findings demonstrate the microglial involvement in developmental synaptic pruning, synaptic plasticity, neuroprotection, neuronal metabolic support and function [3,31,32,33]. The high number of mildly activated microglia and their recruitment around neuronal-glial clusters with spontaneous synaptic activity recently observed in the study on ECM-mimetic peptide hydrogels suggested that the cells are indeed involved in forming such tissue-mimicking assemblies [5]. Activated microglia is typical of the developing brain when neuronal selection and synaptic pruning occur [34,35]. At this stage, microglia secrete nanomolar levels of pro-inflammatory cytokines such as IL-1β and TNFα, which control the functional plasticity [16,36,37]. A more detailed examination of microglial morphology in the present study revealed a high diversity of microglial morphology on the CLP-PEG hydrogels: the cells can be round, flat and spread, rod-shaped branchy-ramified. The varying morphology of the microglial cells was also characteristic of the developing cerebellum [22,38], where they can be round, amoeboid, stout, branched with thick processes containing phagocytic cups, and branched with thin processes microglia observed. Both amoeboid and thick-processed microglia have been described as phagocytic, the stout—as immature, and the ones with the lean processes—matured and neuronal function-supporting. Our time-lapse imaging of microglia revealed that the microglial cells could quickly transform from one shape to another on the CLP-PEG hydrogels. In addition, the motility experiments demonstrated that amoeboid microglia were relocating more rapidly than other forms, suggesting the shape is most suitable for the fast movement and is adopted when the microglial cell is strongly attracted to a specific site. The fact that the microglial cells become round when changing from one form to another suggests that the round shape relates to a transient state between the different phenotypes, a sort of “watching mode”, in which the cells are ready to sense the environmental signals and react by fast action. The microglial cells’ thin processes can also be developed and retracted quickly to respond to environmental signalling. The shape of stout microglia resembles the rod-shaped cells on the CLP-PEG hydrogels. Such bipolar microglia were described as fast-proliferating by Tam and Ma [39]. These findings suggest that microglial cells are highly dynamic, and different morphology reflects their particular activity at the moment of observation rather than their maturation level. For example, round/amoeboid microglia are moving and ready to convert, rod-shaped/stout are proliferating, and branchy-ramified are environment-inspecting/synapse controlling ones. When ramified microglia phagocytose, the ends of their processes become thick and form phagocytic cups.

Surprisingly, we observed no such diversity in the microglial population cultivated on the collagen hydrogel: the microglial cells were round and similar to those in the regular cultures on plastic. Assuming the round shape indeed represents the vigilant state waiting for the signal to respond, all the microglia in collagen hydrogel and plastic-supported cultures were in the waiting mode. One of the possible explanations might be the lack of signalling from other cell types (neurons or astrocytes). Indeed, the neurons did not develop spontaneous Ca^2+^ signalling activity on the collagen hydrogels and plastic during seven days in culture.

An additional explanation for the observed microglial shape and motility differences might be related to differences in the nanostructure of the hydrogel substrates. Although the chemistry of the central constructive units of CLP-PEG and collagen hydrogels is similar, the building blocks of CLP-PEG are significantly smaller, and the overall hydrogel structure is anisotropic. This difference might impact the strength and size of the focal adhesions leading to different scenarios in the cell culture formation. The collagen hydrogel tested consists of type I collagen, with the highest triple helicity (over 96%) compared to the other family members [40]. The collagen hydrogel mainly stimulated astrocytes but poorly supported neurons and microglia. Collagen type I is recently reported to induce neocortex folding by forming invaginations [41]. Collagen IV promoted neural progenitor differentiation and blocked astrocyte differentiation [42]. Thus, different collagen forms have to be taken into account when designing collagen-based hydrogel matrices as they encode specific information to the cells despite their overall chemical similarity.

In addition to the morphological diversity and motility, microglia on the CLP-PEG more frequently generated spontaneous Ca^2+^ signals and had larger signal amplitude than microglia on the collagen hydrogel; at the same time, microglial and neuronal cells on plastic did not show any activity on day 7 in culture [5]. We have previously found that in microglial cells, spontaneous Ca^2+^ signals were sensitive to modulators of the mitochondrial permeability transition pore (mPTP) [24]. Transient openings of mPTP reflect spontaneous bursts of ROS generation (mitoflash) [43], which regulate ATP production by reacting to proton leakage through F_1_F_o_ATP synthase [44]. The more frequent Ca^2+^ signal generation recorded in microglia on CLP-PEG hydrogels could be explained by more active and mature neuronal assemblies formed on the hydrogel substrates. Since microglial development is affected by neuronal-derived TGFβ1 [15], more mature neurons could facilitate microglial development. In addition, microglia react to neuronal activity by shaping synaptic connections [16] and by regulating neuronal activity through ATP sensing [17]. Therefore, the rate of the spontaneous Ca^2+^ signals could reflect the activation and maturation of microglial cells.

Finally, the shape-retaining properties of the chemically crosslinked CLP-PEG hydrogels allow for microfabrication of the desired topographies [45], as demonstrated by prototyping microwells for standardization of the neuroglial spheroids. However, although we have shown the feasibility of the concept, the neuroglial spheres in the wells had no connections. Thus, further improvements are required in the design of the hydrogel microdevices supporting both the standard size of neuroglial clusters and neurite connections between them.

## 5. Conclusions

The synthetic CLP-PEG hydrogel substrate, but not type I collagen hydrogel, promotes the formation of primary cerebellar cell organoids with tissue-like microglial morphology and function, including spontaneous Ca^2+^ signalling in neurons and microglia, as well as the microglial motility. These self-supporting, ECM-mimetic substrates provide a unique platform to study functional brain cells by conventional analysis assays developed for 2D cell cultures on laboratory plastic. Combined with microfabrication, they can pave the way to standardized spheroid formats useful in advanced organ-a-chip and HTS applications.

## 6. Patents

The CLP-PEG hydrogel matrix technology described in this manuscript is disclosed in Ferentis UAB patents: US10273287B2, EP3283510B1.

## Figures and Tables

**Figure 1 biomedicines-10-01023-f001:**
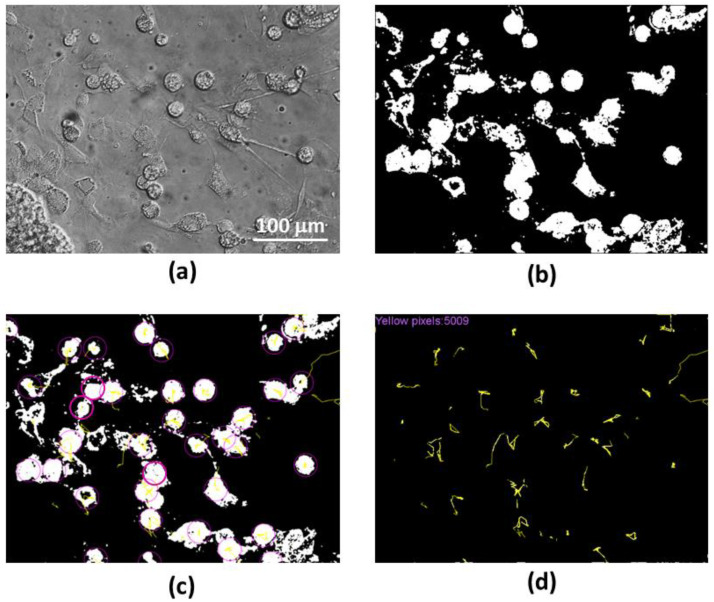
Steps of microglial motility evaluation from time-lapse stacks: (**a**) a representative image of a stack before the assessment; (**b**) the image after conversion to a binary image; (**c**) image was obtained after tracking the selected cells (purple circles) in the stack by the TrackMate; (**d**) the cell tracks and the total number of pixels from all the tracks calculated by the script (in the left upper corner). The scale bar is the same for all images (**a**–**d**).

**Figure 2 biomedicines-10-01023-f002:**
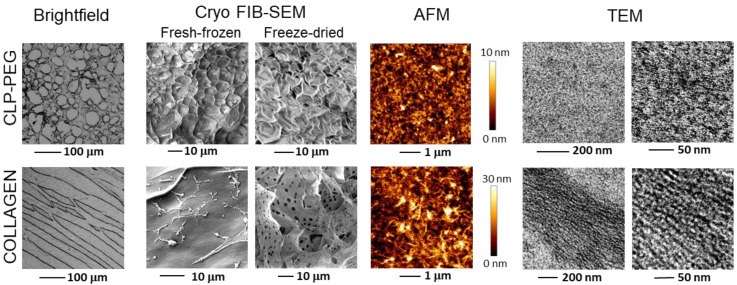
Micro- and nanoscale imaging of the CLP-PEG and collagen hydrogel structure by electron (SEM, TEM) and atomic force microscopy (AFM) techniques.

**Figure 3 biomedicines-10-01023-f003:**
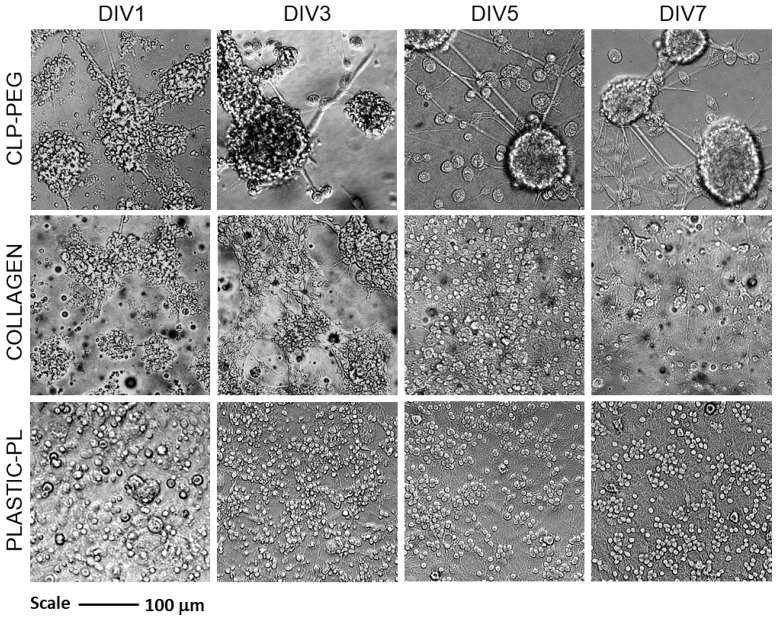
Representative brightfield microscopy images of cerebellar cell cultures on CLP-PEG and collagen hydrogels and on poly-l-lysine (PL) coated tissue culture plastic after 1, 3, 5 and 7 days in vitro (DIV).

**Figure 4 biomedicines-10-01023-f004:**
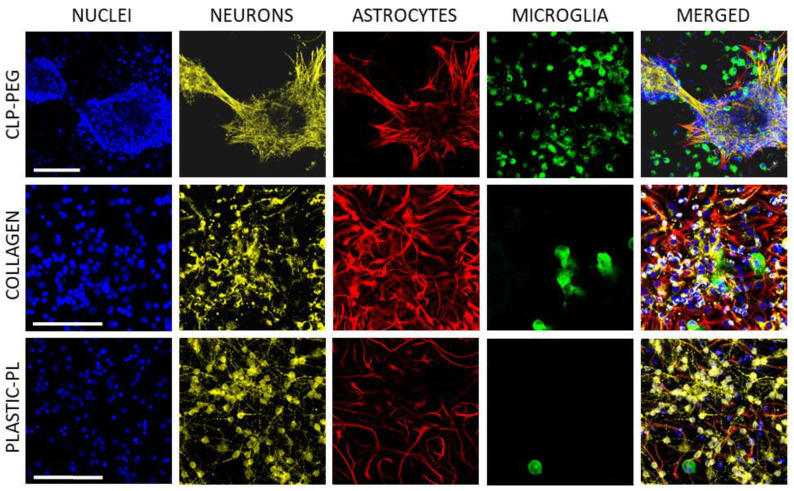
Typical cell culture images of cerebellar cultures after seven days in vitro on CLP-PEG and collagen hydrogels and on poly-l-lysine (PL)-coated glass, respectively. The nuclei were stained blue with Hoechst33342; neurons (yellow) visualized according to microtubule-associated protein 2 (MAP-2), astrocytes (red) according to glial fibrillary acidic protein (GFAP), and microglia visualized green with isolectin GS-IB_4_. The scale bar is 100 μm.

**Figure 5 biomedicines-10-01023-f005:**
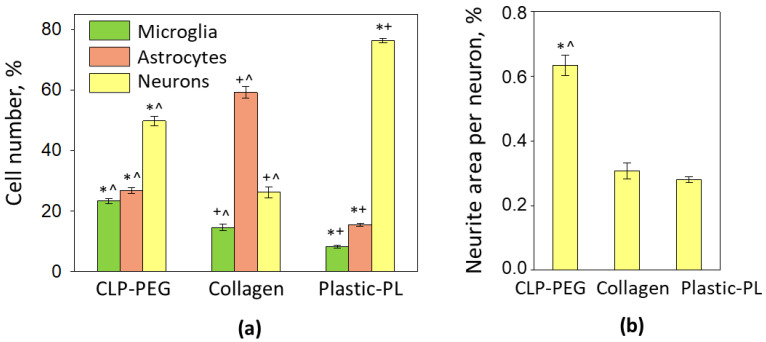
Cellular composition (**a**) and neuritogenesis (**b**) of cerebellar cell cultures on CLP-PEG and collagen hydrogels and poly-l-lysine (PL)-coated tissue culture plastic, respectively. For neurite evaluation, the MAP2-positive area in micrographs was measured by the ImageJ software and expressed as the percentage of the entire image area per single neuronal nucleus. The data are shown as averages of 3–6 experiments with standard error. *—denotes statistically significant difference compared to the collagen; ^—compared to Plastic-PL; +—to CLP-PEG hydrogel; *p* < 0.01.

**Figure 6 biomedicines-10-01023-f006:**
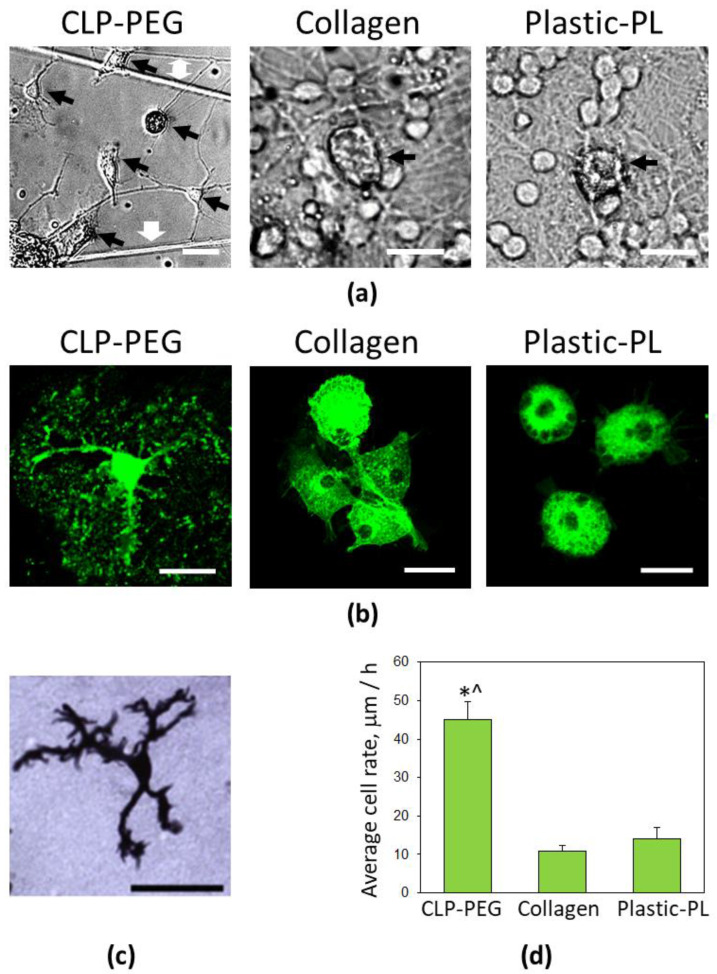
Representative images of microglial morphology and motility on CLP-PEG and collagen hydrogels. In (**a**), phase contrast brightfield images of living microglia in cerebellar cultures on 7th day in vitro. Black arrows point to microglia, white arrows—to neurite fibres. In (**b**), fluorescent microglial images were visualized by isolectin GS-IB_4_ in fixed cerebellar cultures on the 7th day in vitro. In (**c**), an image of a ramified microglial cell in rat cerebellum at postnatal day 12 [22] is presented for comparison, with the kind permission of prof. Margaret M. McCarthy. All scale bars are 25 μm. In (**d**), quantitative evaluation of microglial motility. *—denotes statistically significant difference compared to the collagen samples; ^—compared to plastic-PL samples; *p* < 0.001.

**Figure 7 biomedicines-10-01023-f007:**
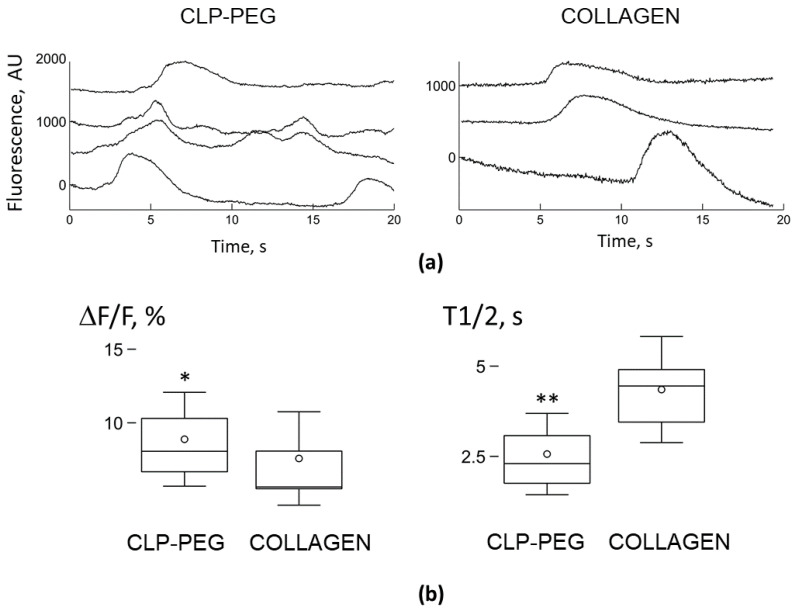
Representative traces of microglial Ca^2+^ concentration changes in cerebellar cell cultures after 7 days in vitro on CLP-PEG and collagen hydrogels (**a**) and quantitative evaluation of the Ca^2+^ signalling properties (**b**). Box-and-whisker plots in (**b**) illustrate the difference in the relative signal amplitude, ΔF/F, and the time of decay to a half-amplitude, T1/2, s, of spontaneous Ca^2+^-sensitive fluorescence signals in microglial cells. The error bars indicate the standard deviation, shoulders of boxes indicate 25–75% intervals, a median of the data is highlighted by a horizontal line, and the mean is indicated by a circle. * *p* < 0.05, ** *p* < 0.01. For ΔF/F, *p* was from the Kolmogorov-Smirnov (K-S) test, for T1/2, *p* was from the Welch’s *t*-test.

**Figure 8 biomedicines-10-01023-f008:**
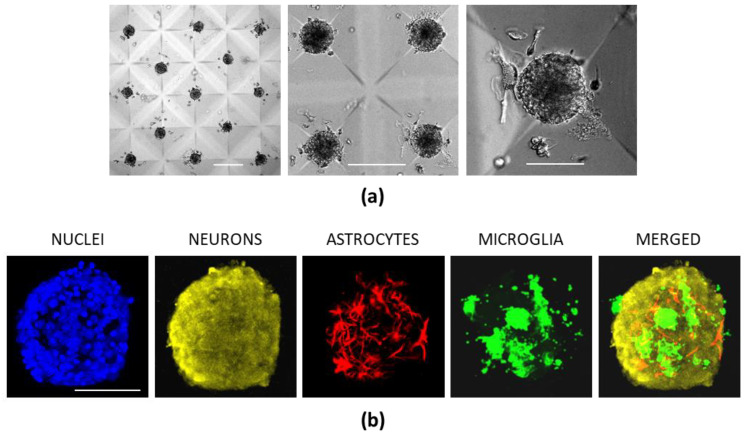
Cerebellar neuronal-glial organoids in microwells fabricated in CLP-PEG hydrogel blocks during seven days in culture. In (**a**), images of living cells in organoids were taken under a brightfield microscope. In (**b**), micrographs of immunostained cells in fixed organoids. Hoechst 33342-stained nuclei are blue, neurons (yellow) are immunolabelled with anti-MAP2; astrocytes (red) with anti-GFAP and microglia are stained green with isolectin GS-IB_4_. The MERGED image is presented without NUCLEI. The scale bar in both (**a**,**b**) panels is 50 μm.

**Figure 9 biomedicines-10-01023-f009:**
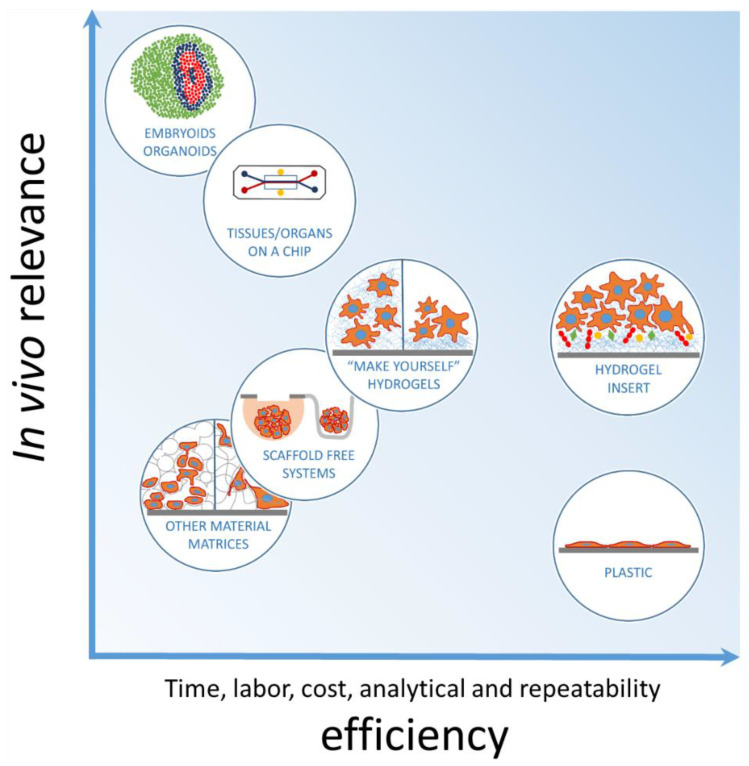
Arbitrary comparison of the common in vitro systems: tissue relevance vs. practical handling and functionality.

## Data Availability

The raw data supporting the conclusions of this manuscript will be made available by the authors, without undue reservation, to any qualified researcher.

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
