# Peer review of "Comparison of Microglial Morphology and Function in Primary Cerebellar Cell Cultures on Collagen and Collagen-Mimetic Hydrogels"

_biomedicines, 2022, doi:10.3390/biomedicines10051023_

Round 1
Reviewer 1 Report
Review of the manuscript which has been submitted to Biomedicines
Manuscript no. biomedicines-1672134
Title: Comparison of Microglial Morphology and Function in Primary Cerebellar Cell Cultures on Collagen and Collagen-Mimetic Hydrogels
I will make this review from the structural point of view and the relevance of the subject. In the current context of the study topic, the article entitled “Comparison of Microglial Morphology and Function in Primary Cerebellar Cell Cultures on Collagen and Collagen-Mimetic Hydrogels” is very interesting and the theme is well chosen, helping the scientific community perhaps reorient certain studies made on neuronal-glial cell cultures development. The results obtained by the authors are impressive and the fact that they obtained cell differentiation on the matrix synthesized by them reflects the quality of the work done in preparing this scientific article. From my point of view, I recommend that the article be accepted for publication in its current form.
Author Response
We thank the Reviewer for evaluating our work and supporting and positive attitude to our study. We are pleased to know that the scientific community finds our research valuable.
Reviewer 2 Report
The work submitted by Z. Balion and coauthors is devoted to such an important issue of regenerative medicine as the development of matrices for the cultivation of brain tissues. In general, the work was performed at a high methodological level, but there are a number of issues that require more detailed explanations.
- To assess the biomimetic characteristics of the scaffold, an evaluation of the physiological activity of all cell types is required. What was the reason for the choice of serum- and glucose-supplemented DMEM medium for the cultivation of neuroglial cultures? As known, DMEM is characterized by greater osmolarity compared to Neurobasal medium traditionally used for obtaining neuroglial cultures, while DMEM is appropriate for pure glial cultures. Glial cells significantly proliferate under these conditions, while the functions of neurons are significantly changed or even suppressed. There are recent works demonstrating normal microglia proliferation in Neurobasal-A medium supplemented with some cytokines.
- Why did the authors choose the 7-day period of cultivation? What happens in these conditions with microglia and neurons at 10-14/21 DIV? It is possible that neurons are non-functional or immature under these conditions (5-7 DIV) and non-synchronous calcium transients or their absence confirm this assumption. To evaluate the neuronal network functionality, bicuculline/picrotoxin or Mg2+-free medium can be added. These exposures in the case of the cortical or hippocampal cultures induce epileptiform activity expressed in mature networks as neuronal quasi-synchronous calcium transients.
- Do astrocytes have a reactive phenotype under the described 2D and 3D cultivation conditions?
- Have the authors evaluated the viability of cells in the experiments?
- Anti-MAP-2 staining in Figure 4 raises some questions. As is known, MAP-2 antibodies stain mainly the processes, as can be seen on the upper panel. However, on the lower panel (and partially on the middle one), a strong staining of bodies is found, which may indicate non-selective staining. It is better to provide a detailed antibody loading protocol, especially the time of exposure to Triton X-100 (this information is absent in reference 5).
Minor points
- Please, add the scale bars in Figure 1.
- I would recommend using “neuroglial cell cultures” instead “neuronal-glial”.
Author Response
Thank you for the revision. Pleas find uploaded the reply to your valuable comments as a pdf file.

Round 2
Reviewer 2 Report
All my comments have been addressed. The authors have provided exhaustive explanations.